# Tracking N- and C-termini of *C. elegans* polycystin-1 reveals their distinct targeting requirements and functions in cilia and extracellular vesicles

Jonathon D. Walsh[1], Juan Wang[1], Molly DeHart[1], Inna A. Nikonorova[1], Jagan Srinivasan[2], Maureen M. Barr[1]*

1 Rutgers, The State University of New Jersey, Department of Genetics and Human Genetics Institute of New Jersey Piscataway, Piscataway, New Jersey, United States of America, 2 Worcester Polytechnic Institute, Department of Biology and Biotechnology, Life Sciences and Bioengineering Center, Gateway Park, Worcester, Massachusetts, United States of America

* mmbarr@rutgers.edu

**Data Availability Statement:** All relevant data are within the manuscript and its Supporting Information files.

## Abstract

The cilium acts as an antenna receiving and sending signals, the latter via extracellular vesicles (EVs). In *C. elegans* and mammals, the Autosomal Dominant Polycystic Kidney Disease (ADPKD) gene products polycystin-1 (PC1) and polycystin-2 (PC2) localize to both cilia and EVs, act in the same genetic pathway, and function in a sensory capacity, suggesting ancient conservation. However, the functions of the polycystins on cilia and EVs remain enigmatic. We used our *C. elegans* model and endogenously fluorescent-tagged LOV-1/polycystin-1 to study LOV-1 processing, trafficking, transport, EV biogenesis, and function in living animals. Super resolution, real time imaging reveals that LOV-1 is processed into N-terminal (NTM) and C-terminal (CTM) forms via a conserved GPCR proteolytic site (GPS). The LOV-1 NTM is secreted into the extracellular matrix and not localized to ciliary tip EVs. In contrast, LOV-1 CTM and PKD-2 are co-trafficked, co-transported, and co-localized in cilia and on environmentally released ciliary EVs. LOV-1 CTM requires PKD-2 for ciliary EV localization, while PKD-2 localizes to ciliary EVs independent of LOV-1. We find that LOV-1 but not PKD-2 is required for chemosensation of an ascaroside mating pheromone. These findings indicate that the polycystins LOV-1 and PKD-2 function together and independently and provide insight to how cargo is selected and packaged in ciliary EVs.

## Author summary

Autosomal dominant polycystic kidney disease (ADPKD) is a common, life-threatening disease that affects 1/400-1/1000 individuals. ADPKD is caused by mutations in PKD1 and PKD2, which encode polycystin-1 and polycystin-2 (PC1 and PC2). Remarkably, the function of the polycystins remains enigmatic almost 30 years after their cloning and 20 years after their discovery on renal primary cilia. Besides cilia, PC1 and PC2 are also found in other subcellular locations including extracellular vesicles (EVs). Urinary EVs

**Funding:** The funders had no role in study design, data collection and analysis, decision to publish, or preparation of the manuscript. Jonathon Walsh, Juan Wang, Inna Nikonorova, and Maureen Barr received salary support from DK116606, DK059418, and NS120745. Jonathon Walsh received salary support from K12 GM093854. Jagan Srinivasan received salary support from DC016058.

**Competing interests:** The authors have declared that no competing interests exist.

can be used as biomarkers of renal disease including ADPKD. Whether these polycystin-carrying EVs are of ciliary origin and what role EVs play in healthy and diseased kidneys remains unknown. In the model organism *C. elegans* and mammals, the polycystins LOV-1/PC1 and PKD-2/PC2 are architecturally similar, act in the same genetic pathway, function in a sensory capacity, localize to primary/sensory cilia, and are shed in EVs, suggesting ancient conservation. Here we use our established *C. elegans* model and fluorescently labeled LOV-1 to study LOV-1 processing, trafficking, transport, EV biogenesis, and function in living animals. We find that the polycystins LOV-1 and PKD-2 function together and independently and provide insight to how cargo is selected and packaged in ciliary EVs.

## Introduction

Autosomal dominant polycystic kidney disease (ADPKD) is one of the most commonly inherited ciliopathies and is caused by mutation in the polycystin-1 (PC1) or polycystin-2 (PC2) gene PKD1 or PKD2 respectively [1–3]. PC1 is an 11-transmembrane domain protein with a large N-terminal extracellular domain that contains a G-protein-coupled receptor (GPCR) autoproteolysis-inducing (GAIN) domain and GPCR proteolytic site (GPS)[4,5]. PC1 is referred to as an atypical aGPCR for its hybrid character of an adhesion GPCR and transient receptor channels [6–8].

Autoproteolytic cleavage of the PC1 GPS domain is important for PC1 function. Mutations that disrupt cleavage are associated with ADPKD in humans and cause kidney cysts in mice [9–11]. Cleavage at the GPS domain of PC1 generates an N-terminal extracellular fragment and a C-terminal transmembrane domain fragment that remain non-covalently associated [9]. The PC1 C-terminal transmembrane domain fragment associates with PC2 and forms the polycystin complex ion channel [12,13]. The N-terminal extracellular fragment is required for some PC1-dependent channel activity and may serve to activate the polycystin complex [14,15]. While the polycystins have likely roles on the endoplasmic reticulum and/or plasma membrane, the primary cilium has emerged as the most likely site of action for PC1 and PC2 [16]. In addition to their ciliary localization, PC1 and PC2 also co-localize on urinary extracellular vesicles (EVs)[17]. The role of the non-covalent association of the N-terminal extracellular fragment and C-terminal transmembrane domain fragment of PC1 and its relationship with PC2 on cilia and EVs remains unclear [9,18]. In this paper, we set out to study *in vivo* PC1 autocleavage in cilia and EV localization in relation to PC2 using the *C. elegans* model.

In "the worm," the polycystins LOV-1/PC1 and PKD-2/PC2 are expressed in male-specific sensory neurons, localize to cilia, and are shed in cilia-derived EVs into the extracellular matrix inside the animal and also into the environment [19–21]. PKD-2 is not a general regulator of EV biogenesis [22]. We used CRISPR/Cas9 mutagenesis to generate LOV-1/PC1 endogenously tagged fluorescent reporters to study LOV-1 subcellular localization, trafficking, interaction with PKD-2, and function. To study the role of the GPS domain, we generated a point mutation in the LOV-1/PC1 endogenously tagged fluorescent reporter strain that is predicted to prevent autoproteolytic cleavage. We discovered physiologically relevant roles for the GPS of LOV-1 in ciliary and EV localization, uncovered different trafficking mechanisms of polycystins, and identified a PKD-2-independent function of LOV-1 in chemosensation. Our results provide new insight to molecular mechanisms underlying polycystin trafficking, localization, and function *in vivo*.

## Results

### N-terminal and C-terminal tagged LOV-1 show different subcellular localization patterns

To study the endogenous LOV-1 protein localization and function of the GPS motif, we CRISPR edited the *lov-1* genomic locus to encode double-tagged reporters on both N- and C-termini (mScarlet::LOV-1::mNeonGreen) (Fig 1A). Endogenous LOV-1 was expressed in male-specific ciliated sensory neurons, similar to published extrachromosomal reporters [19,23,24] (Fig 1B–1G). N-terminal and C-terminal tagged LOV-1 reporters exhibited different localization patterns (Fig 1C–1E). The C-terminally tagged LOV-1::mNG was enriched at the ciliary base and tip and shed in EVs, similar to PKD-2[25]. In contrast, N-terminally tagged mSC::LOV-1 was not enriched at the ciliary tip. Instead, mSC::LOV-1 localized to the ciliary base, shaft, and in the extracellular matrix surrounding the CEM cilium (Fig 1C and 1E). These results suggest that LOV-1 is processed into NTM and CTM regions, similar to mammalian PC1. LOV-1::mNG localized on the ciliary membrane and ciliary EVs while mSC::LOV-1 was secreted in the extracellular matrix around the cilium (Figs 1E and S3).

In ray cilia in the male tail, we observed similar differential localization of LOV-1::mNG and mSC::LOV-1 (Fig 1F–1G). LOV-1::mNG was enriched at the ciliary base and tip of the RnB neurons, whereas mSC::LOV-1 appeared abundantly in the dendrite but absent from the ciliary tip (Fig 1G and 1H). Strikingly, LOV-1::mNG, but not mSC::LOV-1, localized to environmentally released ciliary EVs (Fig 1F–1I).

To determine whether the differential localization of LOV-1::mNG and LOV-1::mSC was influenced by mNeonGreen (mNG) and mScarlet-I (mSC), we swapped the location of the two fluorescent proteins in double-tagged LOV-1. We constructed and analyzed an mNG::LOV-1::mSC CRISPR strain (S1 Fig). The localization pattern of N- and C-terminal reporters was similar between mNG::LOV-1::mSC and mSC::LOV-1::mNG. LOV-1::mNG remained on the cilium and ciliary EVs, while mSC::LOV-1 appeared diffuse in the extracellular space. Combined, these results are consistent with LOV-1 processing.

Polycystin-1 (PC1) and Polycystin-2 (PC2) form a complex and function as an ion channel in mammalian and other vertebrate systems [16,26]. Interactions between LOV-1 and PKD-2 were not explored due to the absence of a full-length, functional LOV-1 reporter. To address this knowledge gap, we built strains expressing N-terminally tagged mSC::LOV-1 and C-terminally tagged LOV-1::mSC crossed into an integrated and functional transgenic PKD-2::GFP strain (Fig 2A and 2B)[24]. N-terminally tagged mSC::LOV-1 demonstrated partial colocalization with PKD-2::GFP. Localization specific to mSC::LOV-1 was observed in the soma, in the dendrites, and surrounding the ciliary base. In addition, only PKD-2::GFP localized to the ciliary tip or cilia-derived EVs. C-terminally tagged LOV-1::mSC and PKD-2::GFP colocalized in the endoplasmic reticulum in neuronal cell bodies [24], in dendritic vesicles, at the ciliary base, along the cilium, at the tip, and in environmentally shed ciliary EVs (Figs 2B, 2C and S4). Dual-channel time-lapse fluorescence microscopy showed co-transport of PKD-2::GFP and C-tagged LOV-1::mSC in RnB dendrites (Fig 2D and S1 Movie). Fluorescence profiling of the RnB cilia showed that enrichments of C-terminally tagged LOV-1::mSC and PKD-2::GFP coincide at the ciliary base and tip (Figs 2E and S2). C-terminally tagged LOV-1::mSC and PKD-2::GFP were released on the same EVs (Fig 2F). Furthermore, the endogenous LOV-1::mSC and the transgenic PKD-2::GFP were released in EVs at a similar abundance (Figs 2F and S4). These data indicate that PKD-2 and C-terminal tagged LOV-1, but not N-terminal tagged LOV-1, are co-trafficked from the ER to cytoplasmic transport vesicles destined for the cilium and ciliary tip-derived EVs [24]. This is the first report of live visualization of Polycystin-1 and Polycystin-2 co-release on ciliary EVs.

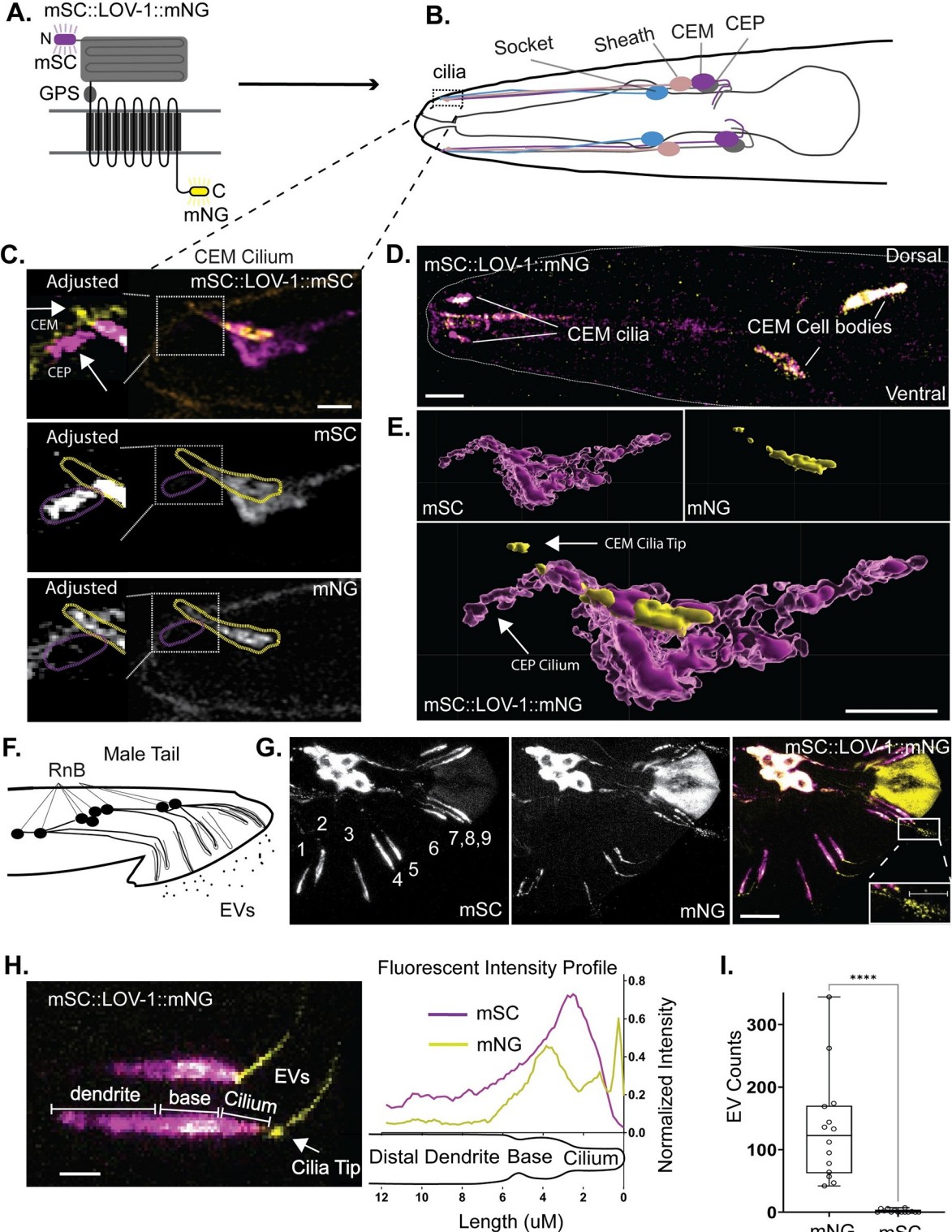

**Fig 1. Double tagged LOV-1 demonstrates different localizations in dendrites and cilia. A)** Cartoon of CRISPR-modified LOV-1 to generate an endogenous fluorescent protein tagged at both the N-terminus and C-terminus with mScarlet-I and mNeonGreen respectively. **B)** Cartoon representing the head of the male worm depicting the CEM, CEPso (socket), and CEPsh (sheath) cells. Dashed box represents the location of the cephalic sensillum that contains the CEM and CEP cilia surrounded by the CEPsh and CEPso glia. **C)** Expression of mSC:: LOV-1::mNG in the male head generated using confocal microscopy. Scale bar = 10 μm. **D)** Maximum intensity projections of z-stack

micrograph of a CEM cilium of mSC::LOV-1::mNG. Left panels were adjusted for brightness to highlight the lower amount of signal surrounding the CEP cilium. Scale bar = 2 μm. **E)** 3D rendering of image in C. C-terminal tagged LOV-1 shows enrichment along the membrane of the CEM cilia (Yellow). N-terminal tagged LOV-1 is enriched along parts of the CEM cilia membrane and is enriched along the membrane of the sheath glia (Dark Magenta). N-terminal tagged LOV-1 is also localized throughout the CEP/sheath lumen and along the CEP cilium at a lower abundance (Light Magenta). Scale bar = 2 μm. **F)** Cartoon representing the male-tail depicting the RnB neurons that express LOV-1 and the EVs that release polycystin-containing EVs. **G)** Confocal microscopy of mSC::LOV-1::mNG in the male tail. Scale bar = 10 μm. Scale bar in EV inset of far-right panel is 1μm. **H)** Fluorescence profiling of the distal region of the ray neurons of mSC::LOV-1:: mNG. Scale bar is 2μm. n = 83. **I)** Quantification of environmentally released EVs from the ray cilia. Statistics performed using Welch's t-test. n = 14.

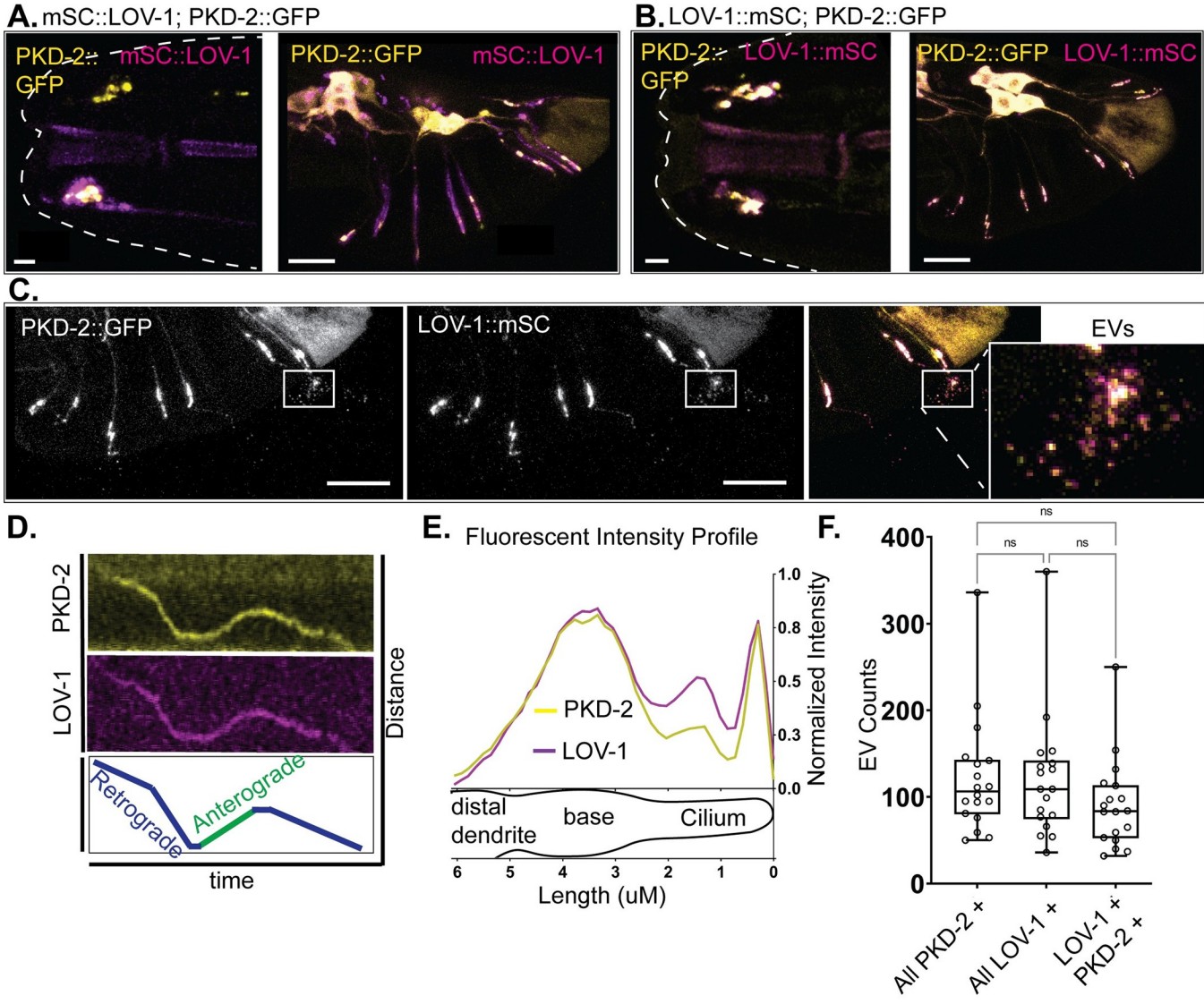

**Fig 2. PKD-2 co-localizes and traffics with CTM LOV-1. A)** Confocal imaging of LOV-1 tagged with mScarlet-I at the N-terminus using CRISPR genome editing in a PKD-2::GFP strain. NTM mSC::LOV-1 demonstrates a broader distribution pattern than that of PKD-2::GFP in dendrites and cilia in the CEM and RnB neurons. Scale bars for head and tail images are 2 μm and 10 μm respectively. **B&C)** Confocal imaging of LOV-1 tagged with mScarlet-I at the C-terminus using CRISPR genome editing in a PKD-2::GFP expressing strain. CTM LOV-1::mSC overlaps completely with PKD-2::GFP. Scale bars for head and tail images are 2 μm and 10 μm respectively. **2C inset)** Both CTM LOV-1::mSC and PKD-2::GFP are released in EVs. Scale bars = 10 μm. **D)** LOV-1::mSC is trafficked together with PKD-2::GFP in dendrites, **E)** overlaps with PKD-2::GFP subcellular localization in cilia (n = 143), and **F)** is co-released with PKD-2:: GFP in the same EVs. n = 18. Statistics performed with Dunn's multiple comparisons.

## GPCR proteolysis site (GPS) is required for localization and function of LOV-1

Polycystin-1 family members are characterized by a GPCR proteolysis site (GPS) at the juxta of the large extracellular N-terminus and the transmembrane domains [8]. Since C- and N-termini of LOV-1 display different subcellular localization, we tested whether GPS autoproteolytic processing was required for the localization of C- and N-termini LOV-1. We created a point mutation in the *mSC::lov-1::mNG* double tag strain to generate an amino acid substitution (C2181S) predicted to prevent proteolytic cleavage due to disruption of a Cysteine-Cysteine bridge necessary for autoproteolysis [9]. Confocal imaging of mSC::LOV-1(C2181S)::mNG demonstrated that LOV-1 localization is dependent on the GPS domain (Fig 3A and 3B). CTM LOV-1 (C2181S)::mNG was completely absent from RnB cilia (Fig 3B). NTM mSC::LOV-1(C2181S) localization was also severely reduced, except in the three ventral ray neurons (R2,4,8B), where NTM mSC::LOV-1(C2181S) was partially enriched (~50% of WT) in distal dendrites and cilia (Figs 3C and S5). These data suggest either incomplete cleavage in the C2181S mutant or other cleavage sites are present that are not disrupted by the C2181S substitution. The latter is most likely as the CTM LOV-1::mNG signal is completely absent away from the soma. Other cleavage sites are observed in human PC1 including an N-terminal site upstream of the GPS in an alternative spliceform (Trunc_PC1) [27]. Additional C-terminal cleavage sites that produce products of various and unique functions are also described, however, the N-terminal variants that result from these other C-terminal cleavages have not been explored [28–35].

To determine how the C2181S mutation affects the LOV-1 protein, we tried to do westerns using two different methods to no avail. First, we did westerns with isolated EVs enriched for PKD-2::GFP using methods described in Nikonorova et al (2022) [36]. We could not detect PKD-2::GFP using highly sensitive GFP antibodies and using super-sensitive ECL reagent. We also tried westerns by picking 1,000 adult males, flash freezing in liquid nitrogen, and crushing in lysis buffer. This was insufficient material to see even with very good mNG nanobody to LOV-1::mNG. In the future, we are considering using 100% male producing strain (new strain using auxin inducible degron to select for males only) [37]. We would use nanotrap beads to immunoprecipitate wild-type and C2181S LOV-1::mNG from lysates of millions of males.

## LOV-1::mNG requires PKD-2 for exiting the cell body and optimal localization to cilia and ciliary EVs

Is PKD-2 required for LOV-1 localization? We examined the localization of double-tagged CRISPR mSC::LOV-1::mNG in a *pkd-2(sy606)* null mutant [19,20] (Fig 3C). Surprisingly, we found that only the C-terminus, but not the N-terminus, of LOV-1 strictly required PKD-2 for exiting the soma (Fig 3C). In *pkd-2(sy606)* males, NTM mSC::LOV-1 is severely reduced in dendrites and cilia of ray neurons, except for the three ventral rays (R2,4,8B). In these ventral ray neurons, the NTM mSC::LOV-1 was enriched (~50% of WT) in the distal dendrites and cilia, similar to the *lov-1*(C2181S) GPS mutant (S5 Fig). The most notable difference between the *pkd-2(sy606)* and *mscarlet::lov-1(*C2181S*)::mneongreen* mutants was reduced cell body abundance of CTM LOV-1::mNG in *pkd-2(sy606)* but not LOV-1(C2181S) (S6 Fig). These data demonstrate that PKD-2 is required for CTM LOV-1::mNG exit from the cell body and suggests that CTM LOV-1::mNG is unstable or is actively degraded in the absence of PKD-2.

## The LOV-1 GPS domain is essential for function in mating behavior

LOV-1 and PKD-2 are required for male mating behavior and act in the same genetic pathways [20]. Here we examined male response behavior to hermaphrodite contact and

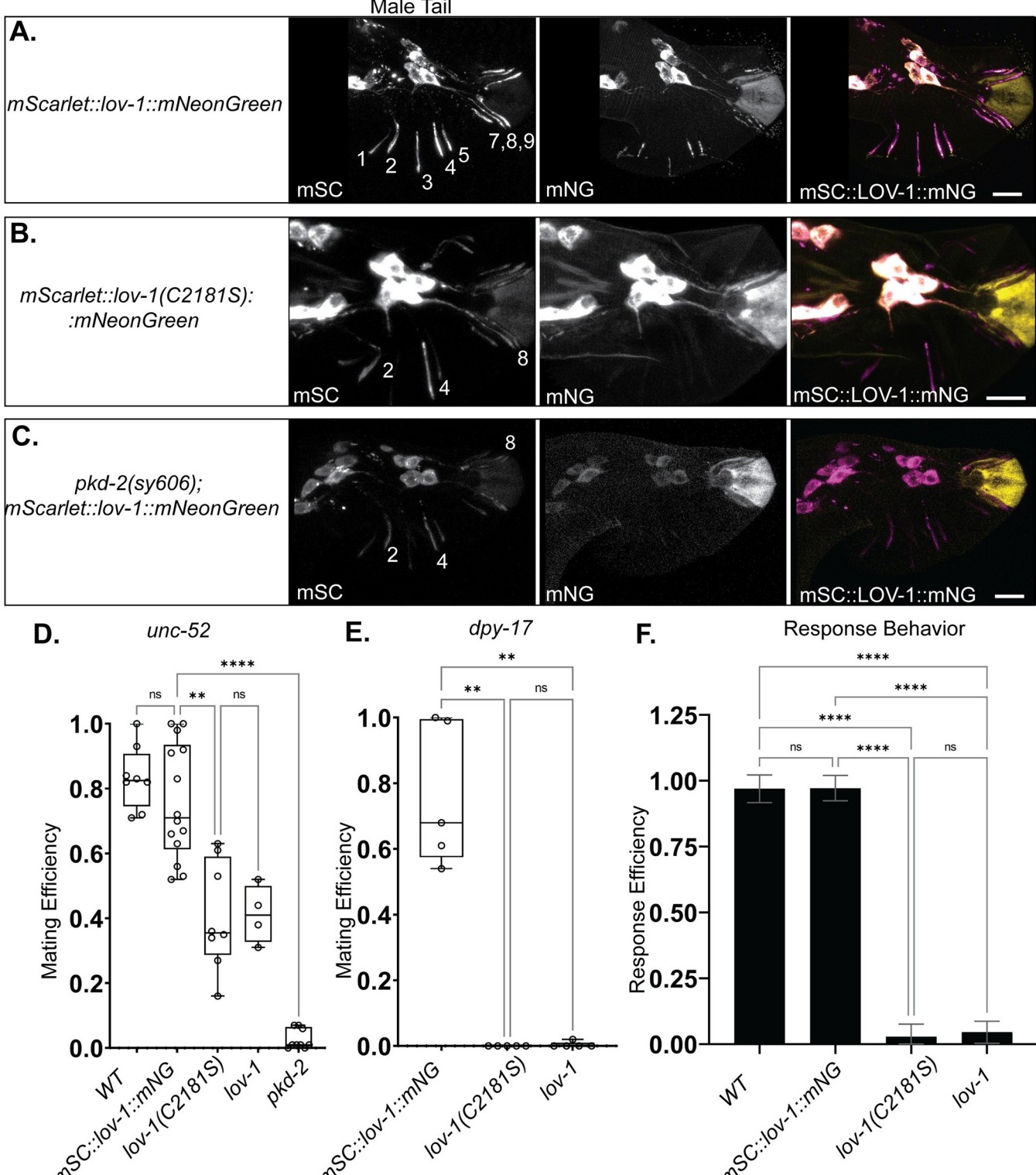

**Fig 3. LOV-1 ciliary localization requires an intact GPS motif and PKD-2. A-C)** Confocal microscopy of mSC::LOV-1::mNG with a point mutation predicted to disrupt GPS cleavage (middle row, C2181S) and a *pkd-2(sy606)* deletion strain (bottom row) compared to WT (top row). NTM mSC::LOV-1 remains enriched in the ventral RnB dendrites in *pkd-2(sy606)* and *lov-1*(C2181S) mutants, whereas CTM LOV-1(C2181S)::mNG is absent from all cilia and dendrites. NTM mSC::LOV-1(C2181S) and CTM LOV-1(C2181S)::mNG remain enriched in the cell bodies of the RnB neurons, whereas both fragments are reduced in the *pkd-2* deletion strain. **D, E)** Mating efficiency assay. **D)** Males were mated with paralyzed *unc-52* or **E)** actively moving *dpy-17* hermaphrodites

to demonstrate degrees of mating difficulties. **F)** Response behavior assay. A WT LOV-1 GPS is required for male mating efficiency and response behavior. n = 36, 64, 63, and 34 for WT, *mSC::lov-1::mNG*, *lov-1*(C2181S), and *lov-1* respectively. **D-F)** Statistics performed using Dunnett's T3 multiple comparisons test. Scale bars = 10 μm.

determined male mating efficiency, the latter measures the number of cross progeny sired by males [38,39]. We examined response behavior of the *lov-1(C2181S)* mutant compared to the deletion mutants of *lov-1(sy582)* and *pkd-2(sy606)* [19, 20]. We confirmed that the double-tagged CRISPR mSC::LOV-1::mNG strain was wild type for male contact response to hermaphrodites (Fig 3D), indicating that the double-tagged LOV-1 in the CRISPR edited strain is functional. In contrast, we found that the GPS-mutated strain mSC::LOV-1(C2181S)::mNG was as defective *lov-1(sy582)* mutants in mating behavior and efficiency assays (Fig 3D–3F).

Two different strains of hermaphrodites were used as mating partners in the mating efficiency assays. The immobile, severely <u>unc</u>oordinated *unc-52* hermaphrodites are easy targets for males while actively moving *dpy-17* hermaphrodites are more difficult mating targets. These *unc-52* and *dpy-17* mating partners discern differing levels of phenotype severity [39]. The GPS mutant *lov-1(C2181S)* displayed mating efficiency defects of differing severity. With *unc-52* hermaphrodites, *lov-1(C2081S)* mutant males displayed a moderate mating efficiency defect, similar to the *lov-1* null allele (Fig 3D). With *dpy-17* hermaphrodites, *lov-1(C2081S)* mutant males displayed a severe mating efficiency defect, similar to *lov-1* and *pkd-2* null animals (Fig 3E). Male response and mating efficiency assays demonstrated that the *mSC::lov-1(C2181S)::mNeongreen* strain fails to initiate and successfully execute mating behavior (Fig 3F) and the defects are not different from the *lov-1(sy582)* deletion allele. We conclude that autoproteolytic processing of the GPS is essential for LOV-1 function.

## Differences in LOV-1 N- and C-terminal fragment dendritic and ciliary transport

The differences in ciliary and EV localization of LOV-1 N- and C-termini lead us to examine dendritic transport and ciliary dynamics of LOV-1. First, we determined whether new LOV-1 fragments are continuously moving into cilia using <u>F</u>luorescence <u>R</u>ecovery <u>after</u> <u>P</u>hotobleaching (FRAP) analysis (Fig 4). FRAP analysis on LOV-1::mNG showed that LOV-1::mNG particle transport stopped at the ciliary base and was continuously enriched at the ciliary base and tip (Fig 4A). The ciliary base signal of LOV-1::mNG recovered to maximum value within 75 seconds (Fig 4A), indicating rapid transport of LOV-1::mNG along the dendrite to the distal most region, the ciliary base (Fig 4A, stick arrow). In contrast, the ciliary tip recovery occurred much slower to maximum value within 250 seconds (Fig 4A, filled arrow). We did not observe moving puncta of LOV-1::mNG along ciliary membrane, which is consistent with diffusion and the absence of visible PKD-2 and GPCR intraflagellar transport [40, 41]. The FRAP analysis on mSC::LOV-1 showed similar rates of recovery between the ciliary base and cilium proper (maximum value within 75 seconds). mSC::LOV-1 did not recover at the cilia tip (Fig 4B). These data indicate that both NTM mSC::LOV-1 and CTM LOV-1::mNG entered the base of cilia where they were unpackaged from dendritic vesicles to the periciliary membrane compartment [24, 42]. CTM LOV-1::mNG moved onto the ciliary membrane where it ended up at the ciliary tip and released in EVs, whereas NTM mSC::LOV-1 diffused freely inside the ciliary base and cilium proper but was restricted from the ciliary tip.

To test the hypothesis that NTM mSC::LOV-1 and CTM LOV-1::mNG are trafficked differently, we measured movement of N- and C-terminal tagged LOV-1 puncta within sensory dendrites using time lapse confocal microscopy. Both N- and C-terminal cleavage products were trafficked bidirectionally (Fig 4C and 4D). The distribution of anterograde velocities of

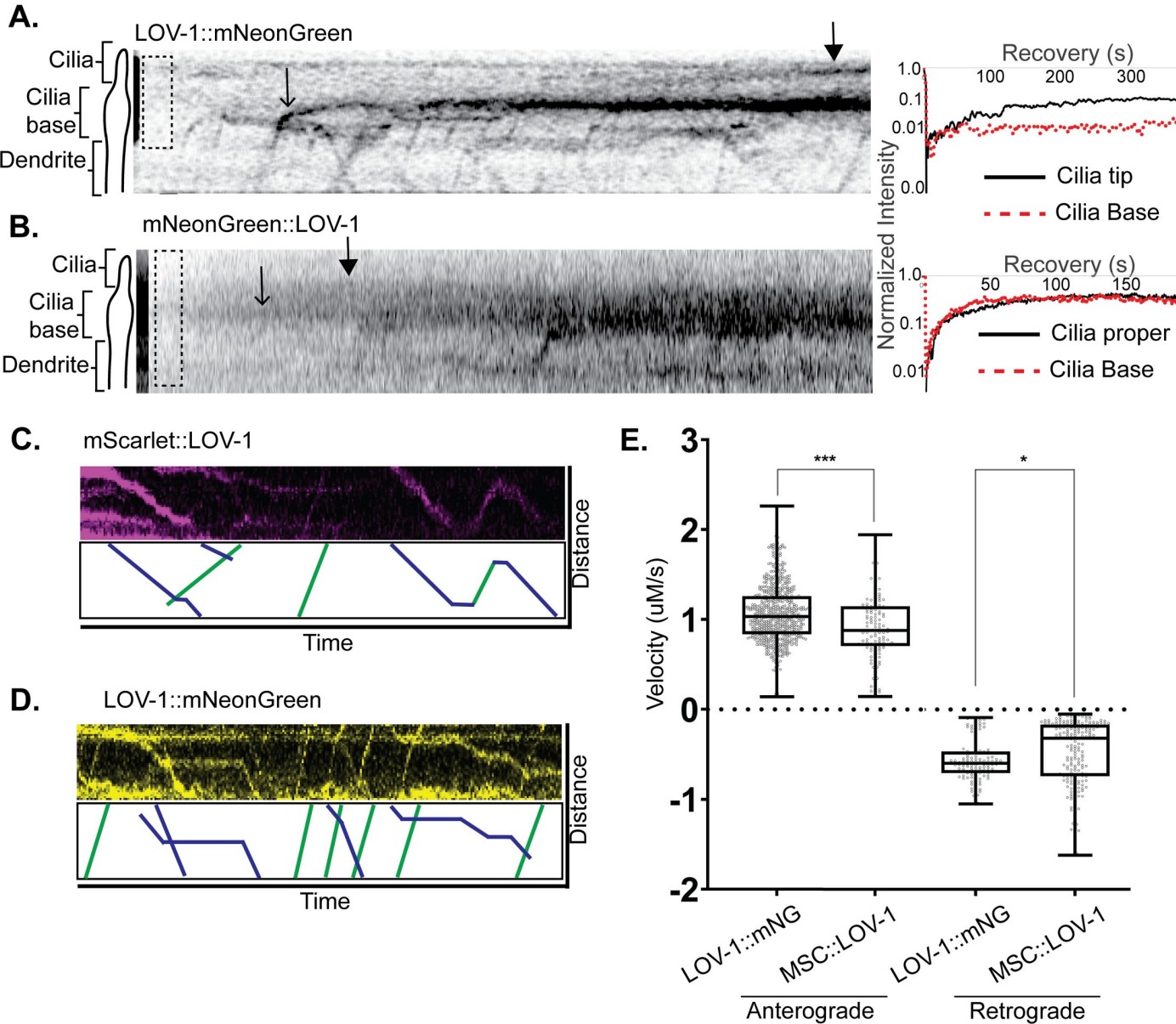

**Fig 4. Ciliary and dendritic trafficking of NTM and CTM LOV-1. A&B)** FRAP experiments demonstrate differences in ciliary trafficking between the CTM and NTM fragments of LOV-1. **A)** Representative kymograph of FRAP of a ray cilium expressing NTM mNG::LOV-1. Dotted box represents the FRAP region. Stick arrow points at the position along the cilia base where there is an accumulation point. Filled Arrow points at a second point of accumulation in the cilium proper. FRAP curve (right) shows successful FRAP. **B)** Representative kymograph of FRAP of a ray cilium expressing CTM LOV-1::mNG. Dotted box represents the FRAP region. Stick arrow points at the position along the cilia base where there is an accumulation point. Filled Arrow points at a second point of accumulation at the ciliary tip. FRAP curve (right) shows successful FRAP. **C&D)** Representative kymographs showing bidirectional trafficking of both **C)** NTM mSC::LOV-1 and **D)** CTM LOV-1::mNG fragments. **E)** Average velocity of anterograde and retrograde trafficking of NTM mSC::LOV-1 and CTM LOV-1::mNG. Statistics performed using unpaired t-tests.

N- and C-terminal LOV-1 fragments were similar, whereas distribution of retrograde velocities were different. CTM LOV-1::mNG moved 0.2–0.3μm/s faster than NTM mSC::LOV-1 (Fig 4E). For CTM LOV-1::mNG, we observed five times the number of anterograde events than retrograde events (S7 Fig and S2 Movie). In contrast, the N-terminal LOV-1 fragment was trafficked two times more often in the retrograde direction (S7 Fig and S3 Movie). These

differences in anterograde and retrograde trafficking between NTM mSC::LOV-1 and CTM LOV-1::mNG further support the hypothesis that differences in trafficking and transport lead to the observed ciliary and EV localization differences.

## LOV-1 and PKD-2 have different dependencies for EV release

To test the relationships between the polycystins in cilia and cilia tip derived EVs, we measured the abundance of polycystin-containing EVs in the context of different polycystin mutants. We found that LOV-1 required PKD-2 and a functional LOV-1 GPS for EV release from ciliary tips (Fig 5A). In contrast, LOV-1 was not required for PKD-2 release in EVs. However, abundance of PKD-2 EVs was significantly reduced in the *lov-1* mutant (Fig 5B). We also observed aggregation of PKD-2::GFP in *lov-1(sy582)* neuronal cell bodies (S9 Fig) as previously described [24]. These data are consistent with the co-trafficking and transport of LOV-1::mNG and PKD-2, indicate that LOV-1::mNG and PKD-2 may function as a signaling EV unit, and that PKD-2 may also function on EVs independent of LOV-1.

## Functional analysis reveals cell-specific roles for LOV-1 and PKD-2

LOV-1 and PKD-2 are required for response behavior which is mediated by RnB ray neurons [20, 43]. To examine the function of LOV-1 and PKD-2 in the cephalic male (CEM) neurons, we used a pheromone sensation assay. Male attraction to hermaphrodites is mediated by hermaphrodite-secreted pheromones (ascarosides) that are detected by CEM and other head neurons [44, 45]. Using an ascaroside choice assay, we performed a comparison of polycystin

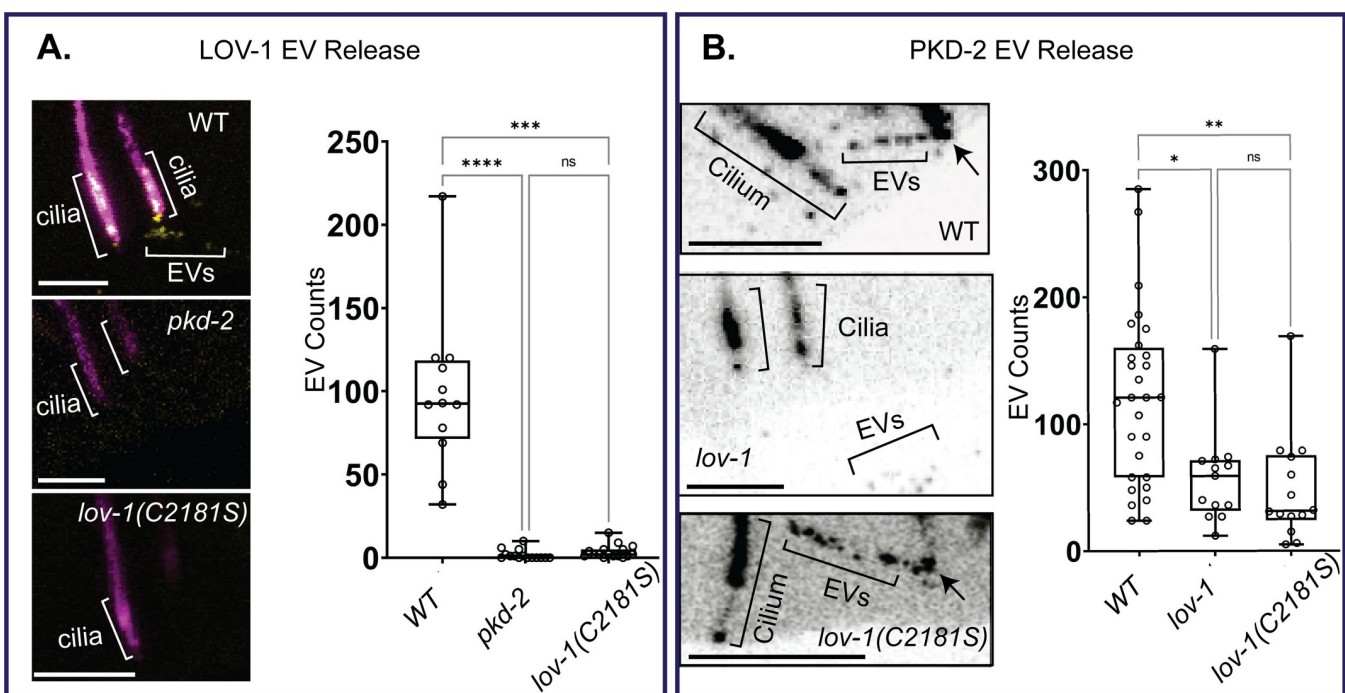

**Fig 5. The C-terminal fragment of LOV-1 requires PKD-2 and a functional GPS for release in EVs, whereas LOV-1 is not essential for PKD-2 release in EVs. A)** Quantification of CTM LOV-1::mNG EV release from *pkd-2Δ* and *lov-1*(C2181S) mutants outside the male tail. Statistics performed using Dunn's multiple comparisons test. n = 13. **B)** Quantification of PKD-2::GFP EV release from *lov-1Δ* and *lov-1*(C2181S) mutants outside the male tail. EVs were visualized by generating sum-slices projections of z-stacks created using confocal microscopy. Statistics performed using Dunn's multiple comparisons test. N = 28, 13, and 14 for WT, *lov-1*, and *lov-1*(C2181S) respectively. All scale bars = 2μm.

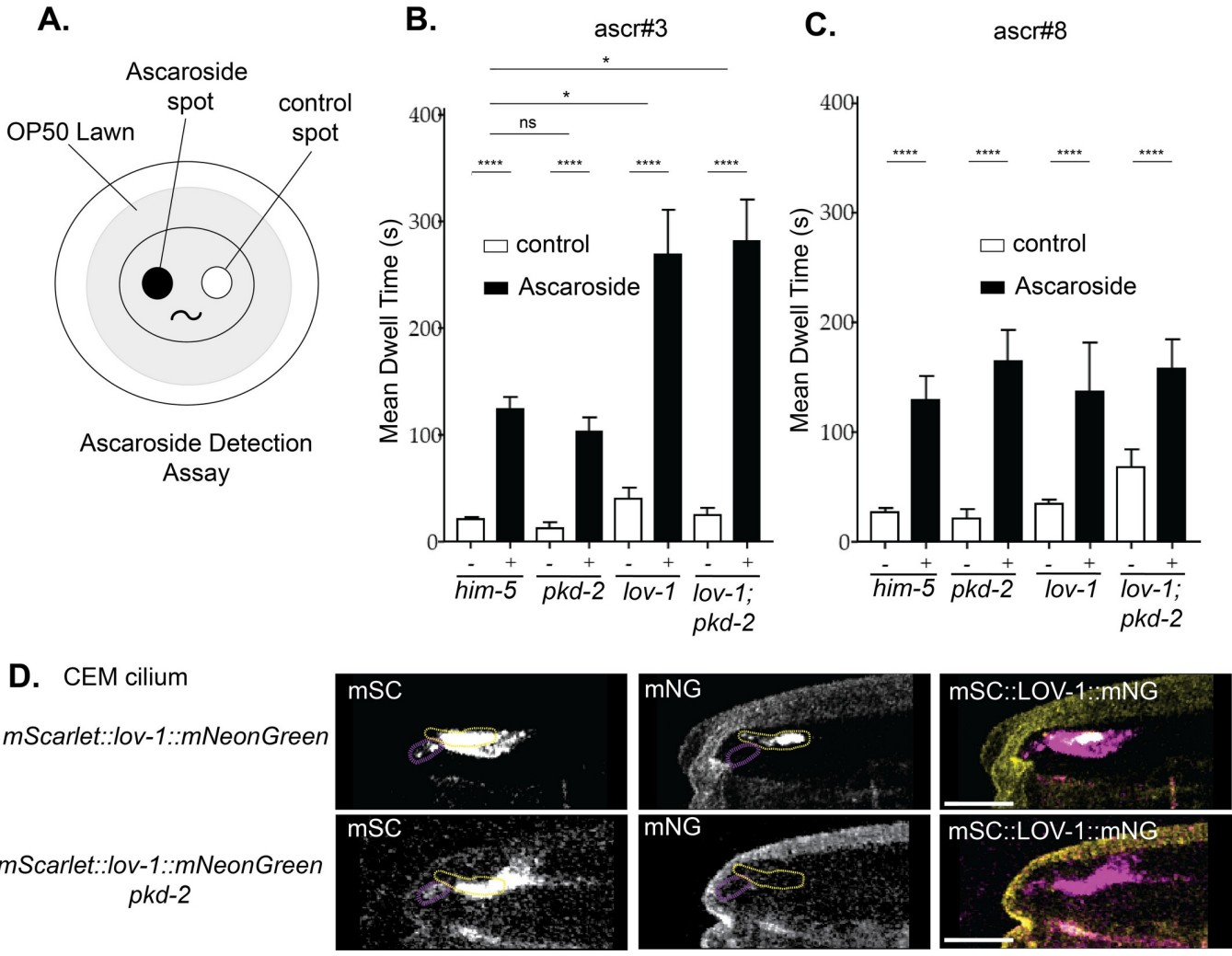

**Fig 6.** *lov-1* **but not** *pkd-2* **plays a role in pheromone detection.** A) Ascaroside detection assay diagram. Largest circle is a culture plate; the gray circle is the food source for the worms. The scoring region consists of two separated droplets. One containing either ascaroside #3 or ascaroside #8 (black circle) the other containing M9 buffer (white circle). **B&C)** Quantification of how long each male placed in the scoring region stayed on the ascaroside spot and the M9 spot in the **B)** experimental and **C)** negative control conditions. High incidence of males *him-5(e1490)* are wild type for mating behavior and efficiency. Statistics performed using unpaired t-test between treatment groups and two-way ANOVA and Tukey's multiple comparisons tests to compare between genotypes. All strains have *him-5(e1490)* in the background. **D)** Confocal microscopy of mSC::LOV-1::mNG in *pkd-2(sy606)* (deletion allele) background. Scale bars = 2μm.

mutants for two different male-attracting ascarosides, ascr#3 and ascr#8 (Fig 6A). *lov-1(sy582)* and *lov-1(sy582); pkd-2(sy606)* males are more sensitive to ascaroside #3, whereas *pkd-2(sy606)* males respond similar to control (Fig 6B). There was no difference between mutants and WT males when exposed to ascr#8 (Fig 6C). We conclude that *lov-1* is a negative regulator of pheromone sensation, and that *lov-1* has a *pkd-2* independent role in ascaroside detection. Interestingly, when comparing localization of the double-tagged fluorescent LOV-1 reporter in CEM neurons of *pkd-2(sy606)* males, CTM LOV-1::mNG was absent from CEM cilia whereas NTM mSC::LOV-1 was still present in CEM cilia and the cephalic lumen (Fig 6D), hinting at a role for NTM LOV-1 in ascaroside detection. These data indicate cell-specific roles for polycystins and provide the first example of a *pkd-2* independent role of *lov-1* in *C. elegans* (Fig 7A and 7B).

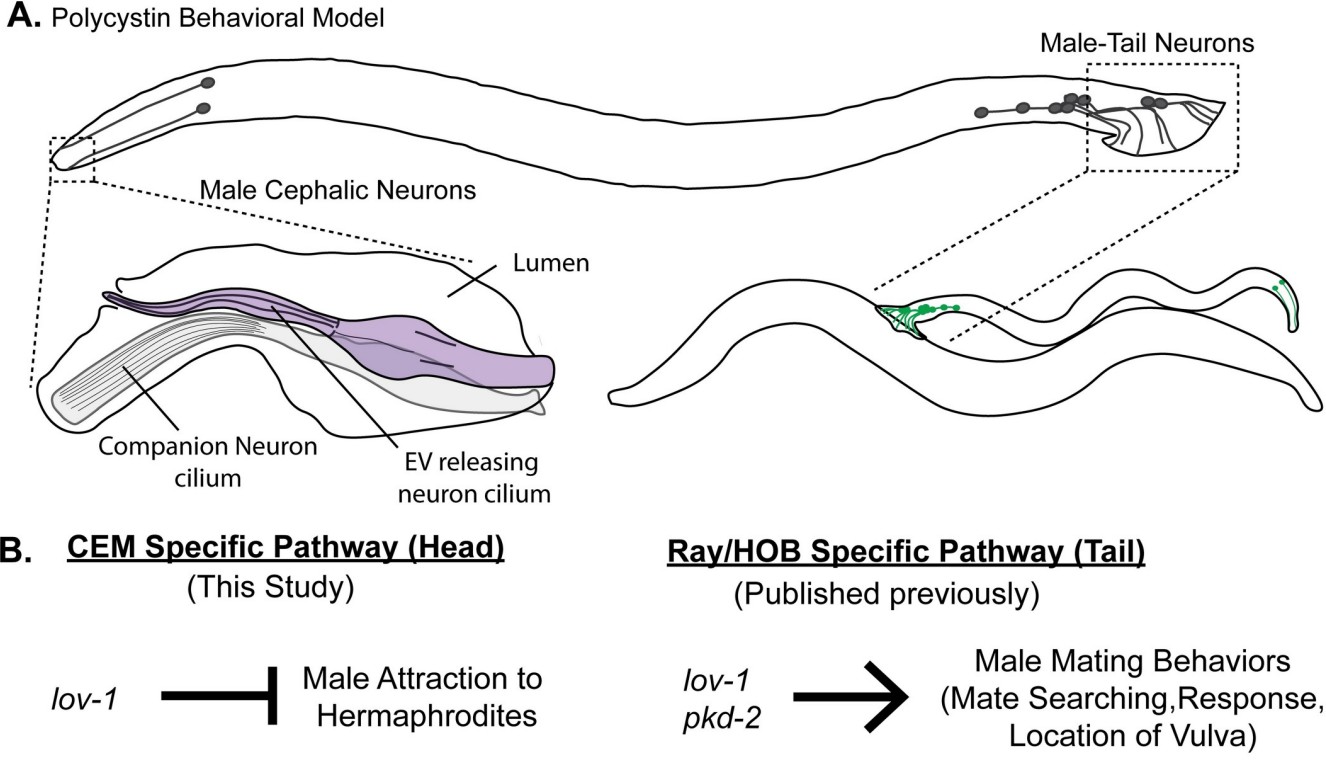

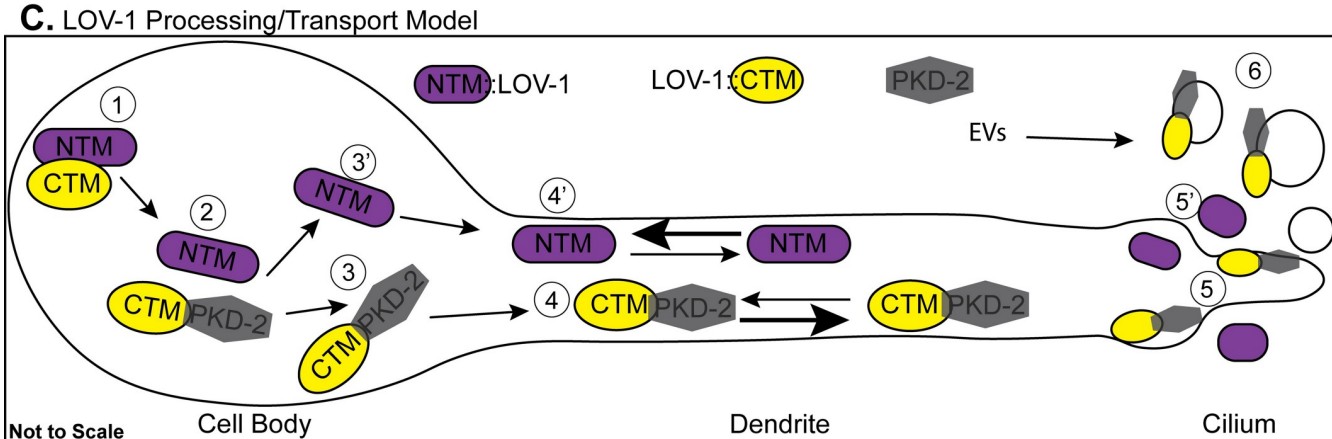

**Fig 7. Models for polycystin function, processing, and transport. A)** Cartoon of the *C. elegans* male and the location of the polycystin expressing neurons (grey) that are responsible for both head-specific and tail-specific behaviors. The cephalic sensillum is comprised of the polycystin expressing CEM cilium, the cilium of the CEP companion neuron, and the lumen created by the surrounding glia cells (sheath and socket). The male tail is a copulatory structure that houses the polycystin-expressing neurons. **B)** Genetic models describing the polycystin pathways in the head and tail that lead to specific behaviors. The *lov-1* dependent CEM-specific pathway mediates male attraction to the pheromone ascaroside #3, whereas both *lov-1* and *pkd-2* act in the tail to regulate response and vulva location behavior. **C)** Model demonstrating GPS processing of LOV-1 and subsequent transport of the polycystins in dendrites. The interaction of unprocessed LOV-1 with PKD-2 (1) is required for proper GPS processing (2). The CTM LOV-1 fragment and PKD-2 remain associated (3) and are co-transported along the dendrite (4), to the cilium (5), and into EVs (6). By contrast, The NTM LOV-1 fragment disassociates (3') and is transported at different anterograde and retrograde rates along the dendrite (4'). The NTM LOV-1 fragment localizes outside of the cilium (5') and is not observed in environmentally-released EVs.

## Discussion

### Polycystin-1 and polycystin-2 colocalize on ciliary EVs and may act as EV signalosomes

The polycystins localize to cilia as well as urinary EVs released from renal epithelial cells [46–49]. The EV functions–if any–of the polycystins are entirely unknown. EVs isolated from ADPKD patients carry different proteins than unaffected individuals [17], and EVs from ADPKD patients promote cyst formation in 3D culture, suggesting that EVs may influence the pathogenesis of ADPKD [50]. PC1, PC2, and the autosomal recessive PKD-gene product fibrocystin colocalize to urinary exosome-like vesicles [49]. The components of exocyst and the ciliary GPCR Smoothened are found in the urinary exosome-like vesicle proteome along with PC1 and PC2[51, 52], suggesting EVs may carry signal transduction modules.

*C. elegans* polycystin-carrying EVs function in animal-to-animal communication. Isolated EVs alter male locomotory behavior and PKD-2::GFP-containing EVs are deposited at the hermaphrodite vulva during mating, suggesting environmentally released EVs are functioning in a signaling capacity [21, 22]. Here, we show that endogenous LOV-1 and PKD-2 colocalize in cilia and on ciliary EVs. These results indicate that polycystin EV biogenesis, targeting, and signaling may be co-regulated. By contrast, other ciliary EV cargoes do not colocalize with PKD-2 on the same ciliary EV even though these cargoes are co-expressed with *lov-1* and *pkd-2* in male-specific sensory neurons [25, 36]. Our EV proteomic analysis identified 2,888 cargo candidates and revealed a heterogeneous EV population. We propose that ciliated cells produce discrete EV signaling units–EV signalosomes–and that the polycystin-carrying EV signalosome mediates inter-organismal communication, in close analogy to inter-cellular signaling in mammals.

The polycystins are co-trafficked, transported, and packaged into ciliary-derived extracellular vesicles (Fig 7C). Our data suggests sorting of the polycystins happens in at least two cellular locations: the cell body and cilia base. In the cell body, most likely the ER based on published PKD-2 localization data [24], the NTM fragment of LOV-1 is separated from CTM LOV-1 fragment and PKD-2 for independent transport in dendritic vesicles. This independent trafficking of the NTM fragment and CTM fragment may be related to endosomal ubiquitin-mediated degradation of the LOV-1-PKD-2 complex [42, 53], which may explain the fewer retrograde transport events of CTM LOV-1 fragment. The NTM LOV-1 fragment may not be subject to ubiquitin-mediated degradation as reflected by more frequent retrograde events comparted to anterograde transport (S7 Fig). At the cilia base, dendritic vesicles carrying the CTM LOV-1 fragment and PKD-2 fuse with the periciliary membrane and gain access to the ciliary membrane where they are targeted to the ciliary tip for EV release. The NTM LOV-1 fragment is also targeted to the cilia base, cilium proper, and secreted into the extracellular matrix by some unknown mechanism (Figs 1 and 4).

### LOV-1 acts in a chemosensory capacity independent of PKD-2

Polycystin family members can act as cellular sensors for perceiving physical and/or chemical stimuli. There is a multitude of evidence that PC1 acts as a mechanosensor [33, 54–57]. On the other hand, polycystin paralogs PKD1L3 and PKD2L1 function as chemosensors in taste buds to sense changes in pH [58–60]. Polycystins can function as primary sensors [61] or as downstream modulators [62]. Here we show that LOV-1 acts in chemosensation independently of PKD-2 (Figs 6 and 7B). The cephalic male CEM neurons are required for modulation of responses to the ascaroside pheromones ascr#3 and ascr#8[63]. Whether LOV-1 is the direct pheromone sensor or acts as a downstream modulator is not known.

This finding of PKD-2 independent function of LOV-1 was surprising. LOV-1 and PKD-2 function in the same genetic pathway in three distinct male mating behaviors: response to mate contact, location of the mate's vulva, and sex drive (Fig 7C)[19, 20, 64]. Consistent with the *pkd-2* independent genetic pathway of pheromone detection, our endogenous LOV-1 CRISPR reporter shows that in a subset of ray neurons in the tail (R2B, R4B, R8B neurons), the N-terminus of LOV-1 localizes to dendrites and cilia independent of PKD-2 (Fig 3). Our observation that ciliary targeting of endogenous LOV-1 is independent of PKD-2 in a subset of the polycystin expressing neurons is consistent with the complexity of cell-specific function of polycystin-1 and polycystin-2 in mammals. We hypothesize that LOV-1 may function in distinct sensory modalities such as chemosensation or mechanosensation in a cell-specific manner.

Interestingly, the four CEM neurons show different responses to ascarosides that are not correlated with their anatomical positions/identities, which seems to be stochastic and is required to create a specific response [63]. While synaptic modulation has been suggested to play a role in this, a possibility might be that LOV-1 is differentially processed and/or localized to individual CEMs to alter their response profiles. Our endogenous reporter-tagged alleles of LOV-1 will allow us to examine this possibility in the future.

## LOV-1 and PKD-2 are the sole *C. elegans* homologues of the ancient polycystin-1 and polycystin-2 family proteins

The polycystins perform diverse functions in different species such as self-incompatibility in tunicates [65–67], sperm activation in sea urchin [68, 69], mechanosensation and predator response in annelids [70], and gamete formation and mechanosensation in the algae *Chlamydomonas* [71, 72]. The *Chlamydomonas* genome does not encode a polycystin-1 family member. *C. elegans* PKD-2 and human polycystin-2 are more similar than LOV-1 and polycystin-1 (S10 and S11 Figs). We propose that polycystin-1 evolved to perform unique and multiple functions. In mammals, polycystin-1 paralogs have diverse functions. In *C. elegans*, LOV-1 acts in at least four distinct sensory behaviors that rely on physical or chemical stimuli. Investigating multi-modalities of LOV-1 will provide insight on how the ancient polycystin-1 functions alone and in conjunction with polycystin-2.

## *C. elegans* is a model for rapid, cost-effective interpretation of ADPKD pathogenic and missense variants of unknown significance (VUS)

*C. elegans* is used to study human VUS in ciliopathies [73, 74] and candidate risk genes in autism spectrum disorders [75, 76], which aids in assessing the pathogenicity of VUS and provides evidence towards reclassification as benign or pathogenic. Here we demonstrated the importance of a conserved residue in the LOV-1 GPS domain, which in humans is classified by the ADPKD variant database [77]. Our *C. elegans* system is a powerful platform to study the effects of pathogenic and VUS mutations on polycystin structure/function/localization. LOV-1 and PC1 share the GAIN/GPS (GPCR autoproteolysis-inducing) domain, 11 transmembrane spanning regions, intracellular PLAT (polycystin lipoxygenase alpha toxin) domain, and extracellular TOP (tetragonal opening of polycystin) domain [19, 78–81]. PKD-2 and PC2 share six transmembrane spanning regions, TOP, voltage sensing and pore domains [19, 20, 82, 83]. Remarkably, the shared TOP domain contains multiple highly conserved residues among polycystin-1 and polycystin-2 that are implicated in human ADPKD (S11 Fig). Further work will focus on generating point mutations in conserved residues of polycystins in *C. elegans* and studying the impact of the mutation on LOV-1 or PKD-2 expression, trafficking, localization, and function.

### The N-terminal fragment of LOV-1 may be regulated by extracellular matrix

Extracellular matrix (ECM) is important for neuronal anatomy and organization of the brain and nervous system [84]. Abnormal ECM and fibrosis are observed in ciliopathies and neuro-degenerative diseases including Alzheimer's disease. Polycystin 1 and its interaction with ECM is implicated in cyst formation [56, 85]. EVs are components of the ECM and EVs themselves may carry ECM proteins and ECM-modifying enzymes [86, 87]. These ECM-loaded EVs may serve as essential modifiers of the existing ECM composition, and thus support signaling between ECM and the surrounding cells. *Chlamydomonas* ciliary EVs carry ECM proteins and ECM-degrading proteases [86–89]. We have evidence for extracellular matrix playing a role in *C. elegans* polycystin signaling and EV release. We found that the *mec-1*, *mec-5*, and *mec-9* genes encoding ECM components regulate PKD-2 and LOV-1 ciliary localization, ciliary EV shedding and neuron-glia interactions [90]. MEC-1 and MEC-9 contain EGF/Kunitz domains and MEC-5 is a unique collagen [91]. One interesting possibility is that the LOV-1 N-terminus interacts with these ECM components to regulate sensory organ homeostasis.

Our results suggest that the N-terminus of LOV-1 may play a PKD-2 independent role in inhibiting the ascr#3 response. In the future, we will express only the N-terminus of LOV-1 in a *lov-1; pkd-2* double mutant background to determine its function in ascaroside detection and mating behaviors. In addition, we will generate strains lacking either the N-terminal fragment or the C-terminal fragment to assess LOV-1 structure-function. In both scenarios, we will also explore the effects of expressing a known dominant negative transgene composed of the first 991 amino acids of LOV-1 [19].

Our observation that the N-terminus of LOV-1 can be targeted to dendrites and cilia in the absence of PKD-2 and without GPS cleavage is important for understanding how polycystin signaling works in the mammalian system. It suggests that mutations in the C-terminus of PKD1 and all PKD2 mutations may give rise to N-terminal polycystin-1 (PC1) fragments that can bind ligands but lack signaling activity or act as a dominant negative. This is consistent with overexpression of a truncated N-terminal fragment of LOV-1 in a WT background causing a dominant negative [19]. Exploring this could be very important for our understanding autosomal polycystic kidney disease, especially given that humans (but not mice) have a truncated N-terminal splice variant that might be ligand binding (Trunc_PC1) [27].

## Materials and methods

### Worm culture

Nematodes were cultured on Nematode Growth Media (NGM) agar plates containing a lawn of OP50 E. coli and incubated at 20°C, as previously described [92]. Strains used are listed in S1 Table.

### CRISPR mutagenesis

For creation of the *lov-1::mneongreen*, *mscarlet-I::lov-1*, *lov-1::mscarlet-I*, *mscarlet-I::lov-1::mneongreen, and mneongreen::lov-1::mscarlet-I* endogenous CRISPR tags, we followed the Mello Lab protocol and used pdsDNA that included a short flexible linker sequence between fluorescent protein tags and the endogenous coding sequence [93]. All N-terminal-tagged *lov-1* constructs were inserted after a 21 amino acid signal peptide to ensure proper processing in the ER [94]. The *lov-1(C2181S)* point mutation was generated using ssODNs as described in [95] to create *mscarlet-I::lov-1(C2181S) and mscarlet-I::lov-1(C2181S)::mneongreen* strains. The guide RNA sequences were designed using CRISPOR [96, 97] and silent mutation sites for

PAM modification and diagnosis were located using WatCut silent mutation scanning [98]. Reagents used are as follows: dg357 was a gift from Dominique A. Glauser and was used in accordance with the *C. elegans* group license with AlleleBiotech. pSEM90 was a gift from Thomas Boulin. All DNA and RNA sequences used are listed in S2 Table. S.p. Cas9 Nuclease V3 (cat#1081058) was purchased from IDT.

### High resolution confocal microscopy

Day 1 adult males isolated from hermaphrodites as L4 larvae were anaesthetized with 10 mM levamisole and mounted on 10% agarose pads for imaging at room temperature. Confocal imaging was performed with a Zeiss LSM 880 inverted microscope with an Airyscan superresolution module using a LSM T-PMT detector and ZenBlack software (Carl Zeiss Microscopy, Oberkochen, Germany). Images were acquired using a 63x/1.4 Oil Plan-Apochromat objective in Airyscan Fast mode and deconvolved using Airyscan processing. Image files were imported into Fiji/ImageJ [99] with the BioFormats Importer plugin processing and analysis. Images were placed in Adobe Illustrator for figure assembly.

### Fluorescence profiling and quantification

Confocal images of the head or tail of male worms were taken using the same parameters for all images analyzed for each test group including the same number of Z-slices and at the same thickness. Maximum intensity projections were created and fluorescent profiles were generated with ZenBlack software. For total fluorescence quantification analysis, image files were imported into Fiji/ImageJ [99] with the BioFormats Importer plugin and z-stacks were processed using Z-Project -> Sum Slices. The line selection tool and MultiMeasure plugin was used to quantify the total fluorescence of the RnB and CEM neurons for each analysis. Total quantification data was normalized to min and max of data sets to allow for comparison of replicates taken on different days [(Data point—min)/(max—min)].

### EV counting

For EV counting of a single channel, image files were imported into Fiji/ImageJ [99] with the BioFormats Importer plugin and z-stacks were processed using Z-Project -> Sum Slices. Area outside the worm was selected using the segmented selection tool and images were converted to 8-bit. Images were segmented using a watershed algorithm (in Fiji/ImageJ: [Image-> adjust-> Threshold] → [Process -> Binary -> Make Binary] → [Process -> Binary -> Watershed]). EVs were then counted using the Analyze Particles tool (Analyze-> Analyze Particles).

For colocalization analysis, the Fiji/ImageJ plugin ComDet V.0.5.5 was used [99, 100]. Sum Slices images were opened in Fiji/ImageJ, area outside the worm was selected using the segmented selection tool and ComDet automatically segmented and completed colocalization analysis based on user input parameters.

### Time-lapse microscopy and kymograph analysis

Day 1 adult males isolated from hermaphrodites as L4 larvae were anaesthetized with 10 mM levamisole and mounted on 4% agarose pads for imaging at room temperature. Confocal time lapse imaging was performed with a Zeiss LSM 880 inverted microscope with an Airyscan superresolution module using a LSM T-PMT detector and ZenBlack software (Carl Zeiss Microscopy, Oberkochen, Germany). Images were acquired using a 63x/1.4 Oil Plan-Apochromat objective in Airyscan Fast mode and deconvolved using Airyscan processing. Kymograph Clear and Kymograph Direct were used to generate and analyze kymographs [101, 102].

## Mating behavior assays

Response efficiency assays were carried out according to Liu and Sternberg [43]. Briefly, 12 unc-31 (e169) young adult hermaphrodites were placed on a 1-cm bacteria lawn on an NGM agar plate. Males were added to the mating plate and observed for 5 min. Response efficiency reflects the percentage of males that successfully responded to hermaphrodite contact within 5 min. Triplicate trials were performed for each line to obtain statistical data. Mating efficiency assays were conducted by placing 6 males and 6 hermaphrodites onto a mating plate for 24 hours. Males were then removed from the plate and hermaphrodites were allowed to lay eggs. After each day, cross progeny and total progeny were counted and the mated hermaphrodites were moved to a new plate. These counts were done over the course of 5 days. Mating efficiencies were calculated as the percentage of cross progeny divided by the number of total progeny. Cross strain hermaphrodites were *dpy-17* and *unc-52*[38]. All mating assays were blinded.

## Ascaroside detection assay

Ascaroside detection assays were carried out as described in Srinivasan et al.[44]. *C. elegans* males were harvested 50–60 worms daily at the fourth larval stage (L4) and stored them segregated by sex at 20˚C overnight to be used as young adults the following day. For the attraction experiments we used 1 µM ascr#3 and ascr#8. Ascaroside detection assay was recorded for 15 minutes at 6 fps as previously described [44].

## Supporting information

**S1 Fig. CRISPR-tagged LOV-1 reporters have similar localization patterns when tagged with different fluorescent proteins at each terminus.** A) Diagram of mSC::LOV-1::mNG. B) Diagram of mNG::LOV-1::mSC. C) 40X tile scan of a whole *C. elegans* male expressing mSC:: LOV-1::mNG. Scale bar is 50 µm. D) Z-projection of male head of worm expressing mSC:: LOV-1::mNG. E) Z-projection of male tail of worm expressing mSC::LOV-1::mNG. Inset shows LOV-1::mNG EVs released from the cilia tips of RnB neuronal cilia. F) 40X tile scan of a whole male worm expressing mNG::LOV-1:: mSC. Scale bar is 50 µm. E) Z-projection of male head of worm expressing mNG::LOV-1::mSC. G) Z-projection of male tail of worm expressing mNG::LOV-1::mSC. Inset shows LOV-1::mSC EVs released from the cilia tips of RnB neuronal cilia. The signal in the green channel at the ciliary tip is autofluorescence of the cuticular pore. If we reduce intensity, auto-fluorescent signal is absent but EVs cannot be imaged. D, E, G, H) Scale bars for head and tail images are 2 µm and 10 µm respectively.
(TIF)

**S2 Fig. All eight polycystin expressing ray (RnB) neurons show the same pattern of NTM (mSC) and CTM (mNG) enrichment at the distal dendrites and cilia when expressing mSC::LOV-1::mNG.** NTM LOV-1 (mSC::LOV-1) is enriched at the distal dendrite, cilia base, and cilia proper but is absent from cilia tips. CTM LOV-1 (LOV-1::mNG) is enriched at the cilia base and cilia tip but not the distal dendrite. n = number of ray neurons measured.
(TIF)

**S3 Fig. NTM mSC::LOV-1 is released outside the CEM cilium.** A) Comparison of the width of fluorescent signal at the base of cilia in mSC::LOV-1::mNG worms compared to a soluble GFP ciliated neuronal marker. Statistics performed was an ANOVA with Tukey post-hoc analysis for multiple comparisons. B) Representative image of *pklp-6*::GFP in the male head. CEM and IL2 cilia are visible. Width measurements were limited to the CEM cilia base region. C) Representative image of mSC::LOV-1::mNG used for signal width measurements. LOV-1::mNG and CEM-expressed soluble GFP correlate in width, thus mSC::LOV-1 must be released outside the CEM

cilium. N = 6, 6, and 5 for LOV-1::mSC, mSC::LOV-1, and soluble GFP respectively. Statistics performed using one-way ANOVA and Tukey's post hoc test. Scale bars are 2 μm.
(TIF)

**S4 Fig. Endogenous LOV-1::mSC and transgenic PKD-2 are released from male tail in EVs outside the worm at the same abundance in single expression strains.** Slightly different EV counts are due to background and/or EV movement during imaging. EV counts were done using an automated counting plugin, ComDet, in ImageJ. Stats generated in JASP and Welch's T-test was performed.
(TIF)

**S5 Fig. *pkd-2* and LOV-1 GPS are required for proper localization of NTM mSC::LOV-1 to ray dendrites and cilia.** Fluorescence profiling of NTM LOV-1 (mScarlet) in ray cilia of worms expressing mSC::LOV-1::mNG in WT, *pkd-2(sy606)*, or *msc::lov-1(c2181s)::mng* backgrounds. Percentages above the graph represent the percent reduction of signal in the mutant strains compared to WT. Rays 2, 4, and 8 (the ventral rays) show an enrichment in the mutant strains when compared to the other rays. Statistics performed using 2-way ANOVA and Tukey's multiple comparisons tests. Sample sizes provided in source data.
(TIF)

**S6 Fig. Cell body abundance of CTM LOV-1 is reduced in *pkd-2(sy606)* when compared to WT and *lov-1(GPS)* strains (arrows).**
(TIF)

**S7 Fig. Transport event per kymograph.** CTM LOV-1::mNG transport occurs more frequently in the anterograde direction, whereas NTM mSC::LOV-1 transport occurs more frequently in the retrograde direction in dendrites. N = 46 and 56 for CTM LOV-1::mNG and NTM mSC::LOV-1 respectively. Statistics performed was unpaired t-test.
(TIF)

**S8 Fig. Histograms showing the velocity distributions of CTM LOV-1::mNG and NTM mSC::LOV-1 in anterograde and retrograde directions in dendrites.**
(TIF)

**S9 Fig. PKD-2 accumulation in cell bodies of ray neurons is observed in *lov-1(sy582)* deletion mutants but not *lov-1(C2181S)* mutants.**
(TIF)

**S10 Fig. Conserved domain alignment of the polycystin 1 family proteins in humans and *C. elegans*.** GPS domain highlighted in blue, PLAT domain in magenta, TOP domain in grey, and the polycystin motif in red. Black arrows indicate conserved residues among all aligned proteins that are associated with ADPKD missense mutations in humans. Red arrows indicate residues that are only conserved among PKD1 and LOV- that are associated with ADPKD missense mutations in humans. The dashed box indicates the conserved Cysteine residue of the GPS that was mutated in this study (C2181S).
(TIF)

**S11 Fig. Conserved domain alignment of the polycystin 2 family proteins in humans and *C. elegans*.** TOP domain in grey and the polycystin motif in red. Black arrows indicate conserved residues among all aligned proteins that are associated with ADPKD missense mutations in humans.
(TIF)

**S12 Fig. Multiple sequence alignment of human PC1, human, PC2, *C. elegans* LOV-1, and *C. elegans* PKD-2 TOP domains.** The polycystin motif is highlighted in red. Black arrows indicate conserved residues among all aligned protein domains that are associated with ADPKD missense mutations in humans. These residues may be mutated in human PC1, PC2, or both.
(TIF)

**S1 Table. Strains used in this study.** All newly generated strains are available upon request.
(TIF)

**S2 Table. DNA and RNA sequences used.**
(DOCX)

**S3 Table. Numerical data for experiments performed in this study (Part 1).**
(XLSX)

**S4 Table. Numerical data for experiments performed in this study (Part 2).** Numerical data for S2 Fig
(XLSX)

**S1 Movie. Dual Channel timelapse of LOV-1::mSC; PKD-2::GFP.** LOV-1::mSC and PKD-2:: GFP are trafficked together (white arrow).
(MOV)

**S2 Movie. Timelapse of CTM LOV-1::mNG.**
(MOV)

**S3 Movie. Timelapse of NTM mSC::LOV-1.**
(MOV)

## Acknowledgments

We are grateful to Gloria Androwski and Helen Ushakov for outstanding technical assistance, Jyothi Shilpa Akella, Kaiden Power, and Christopher Ward for discussions, and members of the Rutgers *C. elegans* community for continued support and sound advice. We also thank WormBase (release WS281) and WormBook that were used daily during this project.

## Author Contributions

**Conceptualization:** Jonathon D. Walsh, Juan Wang, Maureen M. Barr.

**Funding acquisition:** Jonathon D. Walsh, Maureen M. Barr.

**Investigation:** Jonathon D. Walsh, Juan Wang, Molly DeHart, Inna A. Nikonorova, Jagan Srinivasan.

**Resources:** Jonathon D. Walsh, Juan Wang.

**Supervision:** Maureen M. Barr.

**Writing – original draft:** Jonathon D. Walsh, Juan Wang, Inna A. Nikonorova.

**Writing – review & editing:** Jonathon D. Walsh, Jagan Srinivasan, Maureen M. Barr.

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
