## [Decision Letter · Decision Letter 0]

11 Oct 2022

Dear Dr Barr,

Thank you very much for submitting your Research Article entitled 'Tracking N- and C-termini of C. elegans polycystin-1 reveals their distinct targeting requirements and functions in cilia and extracellular vesicles' to PLOS Genetics.

The manuscript was fully evaluated at the editorial level and by independent peer reviewers. The reviewers appreciated the attention to an important problem, but raised some concerns about the current manuscript. Based on the reviews, we will not be able to accept this version of the manuscript, but we would be willing to review a much-revised version. We cannot, of course, promise publication at that time.  The comments from the reviewers are easily addressable and it will provide important information to the field.

If you decide to revise the manuscript for further consideration at PLOS Genetics, please aim to resubmit within the next 60 days, unless it will take extra time to address the concerns of the reviewers, in which case we would appreciate an expected resubmission date by email to plosgenetics@plos.org.

We are sorry that we cannot be more positive about your manuscript at this stage. Please do not hesitate to contact us if you have any concerns or questions.

Yours sincerely,

Susan K. Dutcher

Academic Editor

PLOS Genetics

Gregory P. Copenhaver

Editor-in-Chief

PLOS Genetics

Reviewer's Responses to Questions

**Comments to the Authors:**

Reviewer #1: Tracking N- and C-termini of C. elegans polycystin-1 reveals their distinct targeting requirements and functions in cilia and extracellular vesicles. Walsh et al (PGENETICS-D-22-01051).

This is another excellent paper from the Barr lab focusing on the processing and targeting of the worm homologues of the human autosomal dominant polycystic kidney disease (ADPKD) genes.

The lov-1 and lov-2 genes of C. elegans encode the worm homologues of the PKD1 and PKD2 genes of humans. Mutations in PKD1 and PKD2 are responsible for 85% and 15% of ADPKD cases, respectively. LOV-1 (PKD1) is a large protein that is processed into two pieces by GPS cleavage. The C-terminal portion of the protein spans the membrane 11 times and has good homology to its mammalian relative. The N-terminus is a basically a moderately sized mucin with extensive sites for O-linked glycosylation and apart from the GPS cleavage domain is quite different from the ectodomain of the mammalian polycystin-1 (PC1). LOV-2 is a close homologue of polycystin-2 spanning the membrane 6 times and has extensive channel homology and is likely a calcium or potassium channel subunit.

The paper is dependent on some advanced CRISPR targeting and represents a technical tour de force in that the authors have managed to target the N-- and C-- termini of the LOV-1 gene with fluorescent reporter constructs and then create a non cleavable GPS mutant (C2181S) on this colorful background. These worms have had at least 3 CRISPR modifications for one gene. This in itself deserves a paper in own right. Given the small genomic size of the worm (100Mbp) `off target’ events can ignored.

To summarize the paper, the authors show that the N-terminus and C-terminus of the LOV-1 protein have different targeting fates. The LOV-1 protein co-associates with LOV-2 protein in the ER and undergoes cleavage. LOV-1 C-terminus is dependent on LOV-2 (PKD2) for targeting to the tip of the primary cilium and hence to extracellular vesicles-- indeed in the absence of LOV-2, LOV-1 C-terminus appears to be very unstable. The N-terminal ectodomain has a different fate to that of the C-terminus which is not absolutely dependent on LOV-2 or even GPS cleavage (there may be a cryptic proteolytic event that can substitute for GPS cleavage). The N-terminus is ultimately shed into the sheaf surrounding the primary cilium and does not accumulate on the tip of the cilium or enter the ECVs.

GPS cleavage is absolutely necessary for the targeting of C-terminal 11 TM portion of the protein out of the ER and into dendrites/cilia and for efficient tracking of the N-terminus to the dendrites and cilia.

Importantly, the authors show that the fluorescent protein tagged LOV-1 variants are WT for mating function. LOV-1 was not required from extracellular release of LOV-2 positive ECV, but C-terminal LOV-1 release on LOV-2+ ECV was dependent on both GPS cleavage and LOV-2. The authors also show that the lov-1 gene has a new function that is not shared by lov-2. It appears that functional lov-1 has an antagonistic effect on worm response to ascaroside #3 but not ascaroside #8 sensing.

The observation that the N-terminus of LOV-1 is stable and can be appropriately targeted in the context of mutations in the GPS domain and the absence of functional LOV-2 protein is important from a mammalian PKD biologists point of view. It suggests that 3’ mutations in the C-terminus of PKD1 and all PKD2 mutations may give rise to N-terminal ectodomain that may have the ability to bind ligands but lack signaling activity – a dominant negative scenario. This could be very important for our understanding ADPKD (indeed humans (but not mice) naturally make an N-terminal truncation product that might be ligand binding -- Trunc_PC1).

Finally, I believe this is an excellent piece of work and should be published as is. I have found a few typos.

Line 96 aGPCR – a GPCR

Line 113 “the worm,” --”the worm”,

Line 274 neurons is the LOV-1 – neurons in the LOV-1

Reviewer #2: In this nice paper, Walsh and co-workers analyze the processing and trafficking of the LOV-1 PC1 and PKD-2 PC2 polycystins in male sensory neurons in C. elegans. Using endogenously reporter tagged alleles, they convincingly show that the processed N- and C-termini of LOV-1 are trafficked and localized differentially. Trafficking of the C-terminus of LOV-1 requires PKD-2, and the N-terminus of LOV-1 may modulate pheromone sensation independently of PKD-2.

The experiments described in the work are well done and nicely quantified. One of the most interesting findings in the paper is the PKD-2-independent role of LOV-1 in ascaroside sensation and I have a couple of suggestions for additional experiments/discussion to bolster these findings.

1. The experiments suggest that the N-terminus of LOV-1 may play a PKD-2 independent role in inhibiting the ascr#3 response. Is overexpression of just the N-terminus of LOV-1 in a lov-1; pkd-2 double mutant sufficient to decrease their ascr#3 response?

2. Could the authors speculate a bit on the interesting ECM localization of the LOV-1 N-terminal domain?

3. Narayan et al (PNAS 2016) previously showed that the 4 CEM neurons show different responses to ascarosides that are not correlated with their anatomical positions/identities. While synaptic modulation has been suggested to play a role in this, an interesting possibility might be that LOV-1 is differentially processed and/or localized to individual CEMs to alter their response profiles. Having an endogenous reporter-tagged allele of LOV-1 should allow the authors to examine this possibility and would be a strong addition to the paper.

4. GPS disruption - Can the authors comment on if they suspect that cleavage is completely blocked or just attenuated in this mutant from their imaging? If not, could Westerns be useful here? It looks like there are some regions that are more enriched for N- vs C-terminus in Figure 3B. Is the mutant protein just unstable, here also westerns might be useful in simultaneously visualizing the cleavage and levels.

Minor:

1. Please define NTM and CTM in the abstract

2. “is” missing from last sentence in abstract

3. Figure 4A: graph labels are too small and are unreadable

Reviewer #3: In their manuscript, “Tracking N- and C-termini of C. elegans polycystin-1 reveals their distinct targeting requirements and functions in cilia and extraciliary vesicles” Walsh et al. investigate the processing, localization, trafficking and function of polycystin-1 by fluorescently tagging the endogenous protein in worms. Despite intense interest in the polycystins due to their involvement in polycystic kidney disease, detecting endogenous protein and its processed forms has flummoxed the field. This study targets distinct N-terminal and C-terminal fluorescent tags to the endogenous lov-1 locus resulting in doubly tagged, functional polycystin-1 protein expression that functions like wild type indicating that the tags do not disrupt endogenous protein function. While autoproteolytic cleavage of PC1 is critical for its function, this study breaks new ground as it follows the localization, traffic and function of the distinct N-terminal and C-terminal cleavage products and sheds new light on the trafficking events leading to polycystins getting packaged into extracellular vesicles. Using the power of C. elegans genetics, the authors are able to study the PC1 cleavage products in relation to polycytin-2 and show that the C-terminal tagged PC1 fragment is co-regulated in its localization trafficking and EV release with PC2. Furthermore, the authors define PC2 dependent and independent roles of the C-terminal tagged PC1 fragment. In particular the PC2-independent function was unexpected and demonstrates the power of the author’s approach in revealing unsuspected biology. This work is significant in providing the field fundamental information regarding processed PC1 protein and of broad interest to researchers interested in cilia-associate diseases.

Strengths of the manuscript include the in vivo approach coupled to careful imaging enabling direct observation at unprecedented resolution, the generally well-controlled experiments and the discovery of clear functional distinctions between the cleavage products of PC1 and the distinct relation of each to PC2. A few claims are not substantiated or sufficiently explained and should be addressed as follows:

1) In Figure 2, the authors claim that the scarlet and GFP fluors colocalize to the ER but there is not ER marker shown to confirm that localization.

2) Figure 2 and the associated text needs clarification. The figure shows the NTM which is not discussed in the text but is mentioned in the figure legend as being distinct from the CTM- much as Figure 1 just showed. What’s the NTM experiment testing? Furthermore, while the gap of no full length functional LOV-1 reporter previously being available is clear, it is less clear why the LOV-1::mSC construct would label full length protein (or whether it has) since the GPS is presumably intact enabling cleavage. Is PC1 thought to be incompletely cleaved? What am I missing?

3) Perhaps for the same reason, I am also confused in the C2181S mutant experiment. Is there complete cleavage? The authors state that the mutation is predicted to prevent cleavage so if that is then the prediction is that the tags should co-localize. They don’t so is this disrupting cleavage as predicted? A western probed for each of the tags would make interpretation of these data possible.

4) The authors jump around among many distinct statistical tests and it is not clear such a variety is needed: Welch’s, Dunn’s multiple comparison, Dunnett’s T3 multiple comparison, unpaired t-test, one-way ANOVA. Why? Usually Welsh’s is used for data with uneven distribution which this does not appear to be. It is just unclear that so many distinct test are needed especially when the same measurement is being made (eg fig 1 and 5 EV counts). What’s the reasoning behind each test’s selection?

Attention should be paid to:

1) The nomenclature the authors use switches around between the text figures and legends from the actual constructs (mSC::LOV-1::mNG) to shortcuts CTM:LOV-1 (confusing as it doesn’t really reflect that the tag is on C-term) to CTM (which tag?). The nomenclature in worms is clear and consistency is always easier to follow. On a related point, I am not sure the NTF/CTF is helpful as it is barely used since the authors only follow the tagged cleaved products. Could just be spelled out when used and avoid more alphabet soup.

2) Figure 3 (along with legend and associated text) would be easier to follow if the C2181S mutation were used when referring to the mutant and not GPS (which is not a mutation)

3) Swapping the tags to test whether either impacts function is a nice control but I think all the labels in SFig1 are incorrect, along with the legend. Additionally, a diagram like Fig 1A showing the swap and parallel figures to 1 C, G and H would make SupFig1 much easier to follow (because the reader would have just seen it in Fig 1).

4) As often happens some of the figures have been modified since the legends were written. A careful, detailed review is needed so that the reader can follow the story. One example in Figure 1 legend is that the adjusted panels on the left are listed under D when they are displayed in panel C. Similarly, the text sometimes does not appear to refer to the correct panel. One example is line 168 in which the EVs are visible in Figure 2B (not referred to in text here).

5) In Figure 7 for clarity, the authors should consider either swapping steps 6 and 5’ (so EV release on bottom and NTM fragment outside cilia on top) OR putting the whole prime pathway along the bottom and the main (non-prime) pathway along the top. It is just not intuitive that 5’ specifically follows 4’ and that 5, 6 follow 4 at present. I got there but it took several readings of legend and text.

**Have all data underlying the figures and results presented in the manuscript been provided?**

Reviewer #1: **No: **I didnt see the videos

Reviewer #2: None

Reviewer #3: Yes

PLOS authors have the option to publish the peer review history of their article (what does this mean?). If published, this will include your full peer review and any attached files.

Reviewer #1: No

Reviewer #2: **Yes: **Piali Sengupta

Reviewer #3: No

---

## [Decision Letter · Decision Letter 1]

7 Dec 2022

Dear Dr Barr,

We are pleased to inform you that your manuscript entitled "Tracking N- and C-termini of C. elegans polycystin-1 reveals their distinct targeting requirements and functions in cilia and extracellular vesicles" has been editorially accepted for publication in PLOS Genetics. Congratulations!

Yours sincerely,

Susan K. Dutcher

Academic Editor

PLOS Genetics

Gregory P. Copenhaver

Editor-in-Chief

PLOS Genetics

Comments from the reviewers (if applicable):

Reviewer's Responses to Questions

**Comments to the Authors:**

Reviewer #3: The author's have satisfactorily addressed my concerns. Honestly, I don't love the long explanation of the unsuccessful westerns in the text but that is preference, not an issue with the data and conclusions.

**Have all data underlying the figures and results presented in the manuscript been provided?**

Reviewer #3: Yes

PLOS authors have the option to publish the peer review history of their article (what does this mean?). If published, this will include your full peer review and any attached files.

Reviewer #3: No

**Data Deposition**

http://datadryad.org/submit?journalID=pgenetics&manu=PGENETICS-D-22-01051R1

**Press Queries**

---

## [Editor Report · Acceptance letter]

22 Dec 2022

PGENETICS-D-22-01051R1 

Tracking N- and C-termini of C. elegans polycystin-1 reveals their distinct targeting requirements and functions in cilia and extracellular vesicles 

Dear Dr Barr, 

We are pleased to inform you that your manuscript entitled "Tracking N- and C-termini of C. elegans polycystin-1 reveals their distinct targeting requirements and functions in cilia and extracellular vesicles" has been formally accepted for publication in PLOS Genetics! Your manuscript is now with our production department and you will be notified of the publication date in due course.

With kind regards,

Zsuzsanna Gémesi

PLOS Genetics

On behalf of:
